# Three-Dimensional Dual-Mesh Inversions for Sparse Surface-to-Borehole TEM Data

Luyuan Wang, Yunhe Liu , Changchun Yin *, Yang Su, Xiuyan Ren and Bo Zhang

College of Geo-Exploration Sciences and Technology, Jilin University, Changchun 130026, China
* Correspondence: yinchangchun@jlu.edu.cn

**Abstract:** The surface-to-borehole transient electromagnetic (SBTEM) method can provide images at higher resolution for deep earth because its receivers are close to targets. However, as usually the boreholes distribute sparsely, the limited EM data can result in an "equivalent trap" in SBTEM inversions, i.e., the data are well-fitted, but the model is not properly recovered. To overcome this non-unique problem, we propose a dual-mesh three-dimensional (3D) SBTEM inversion scheme. We first use a coarse mesh to obtain a rough resistivity distribution near the borehole, and then we map the coarse mesh attribute into a fine one and capture details from the fine mesh inversion. We test our method on both synthetic data and survey data acquired in Daye, Hubei Province, China. Numerical experiments show that our dual-mesh inversion strategy can better recover the location and resistivity of targets.

**Keywords:** surface-to-borehole transient EM; 3D inversion; dual mesh; non-uniqueness

## 1. Introduction

The surface-to-borehole transient electromagnetic (SBTEM) method uses a grounded wire as transmitter, while the receivers are laid in the borehole to measure EM fields [1]. Since the borehole receivers are close to the targets, the anomaly signal is much stronger than that in the conventional ground EM, so the deep targets concealed around the borehole or at the shaft bottom can be detected [2]. In past decades, the SBTEM method has been widely used in mineral explorations. It has played an important role in the detection of massive sulfide, pyrite-pyrrhotite ore, and gold deposits and in the exploration of deep targets like volcanic veins [2–7].

Although the SBTEM equipment has now been well-developed, the interpretation of SBTEM data is still underdeveloped, usually assuming a simple half-space or layered-earth model. The mainstream imaging methods simply transform TEM data into some intermediate parameters to predict the rough structures in the underground [8], which cannot guarantee that the recovered model can fit the observed data. Zhang and Xiao [9] proposed a method that runs joint inversions of surface and borehole data to one-dimensional (1D) models and found that they can well-detect both the shallow and deep structures. Chen et al. [10,11] successfully conducted the joint inversions and 1D Occam's inversions of SBTEM data. However, all these imaging and 1D inversions have difficulty in recovering the complex underground structures, so the development of three-dimensional (3D) inversions for SBTEM data is necessary.

Since, until now, little has been reported on 3D SBTEM inversions, we review the mainstream 3D forward modeling and inversions of conventional TEM methods. At present, the mainstream numerical simulations for TEM methods include the integral equation (IE) [12,13] the finite-difference (FD) [14], and the finite-element (FE) methods [15,16]. As for the TEM inversions, most conventional 3D algorithms are based on structured grids [17,18]. For accurate modeling and inversions, one needs very small grids to fit the undulating topography or complex underground structures. This can result in heavy computational cost.

To solve this problem, we take in this study the locally refined tetrahedral grids to discretize the computational domain and use the vector FE method to calculate EM responses and sensitivity information. For the time discretization, we choose the unconditionally stable backward Euler method [16,19–21]. Furthermore, we apply the L-BFGS (limited-memory BFGS) method for model updates at two inversion stages. In the SBTEM method, we usually have only sparsely distributed borehole data. To obtain the fine structures close to the borehole, however, the grids around the borehole receivers are generally refined; thus, we actually deal with an inverse problem with sparse data for the fine model. In this case, the borehole data can be easily fitted by a combination of resistivities of fine grids close to the receivers as they usually have large sensitivities. This can bring serious non-uniqueness to 3D EM inversions. To deal with this problem, we propose an inversion strategy for SBTEM that adds the surface EM data to the inversion and applies a dual-mesh scheme. Different from other multi-mesh inversion methods [22,23], we use a coarse mesh to do the first round of inversion to invert the distribution of the background structures, and then, we keep the model and switch to a fine mesh to recover model details and to fit the data.

In the sequence, we first introduce the SBTEM inversion algorithm and then present in detail the dual-mesh inversion strategy. We test our 3D SBTEM inversion scheme via both synthetic and field survey data.

## 2. Methods

As shown in Figure 1, in the SBTEM method, the transmitter placed at the earth surface transmits the EM waves into the earth and creates the eddy currents in the underground. The eddy currents will be distorted when they travel to the targets with anomalous conductivities, so that anomalies will be observed by the sensors placed at the borehole. From these anomalies, we can infer the anomalous bodies. Since the receivers are generally laid close to the anomalous bodies, one can observe large anomalous responses and, thus, can easily detect the targets.

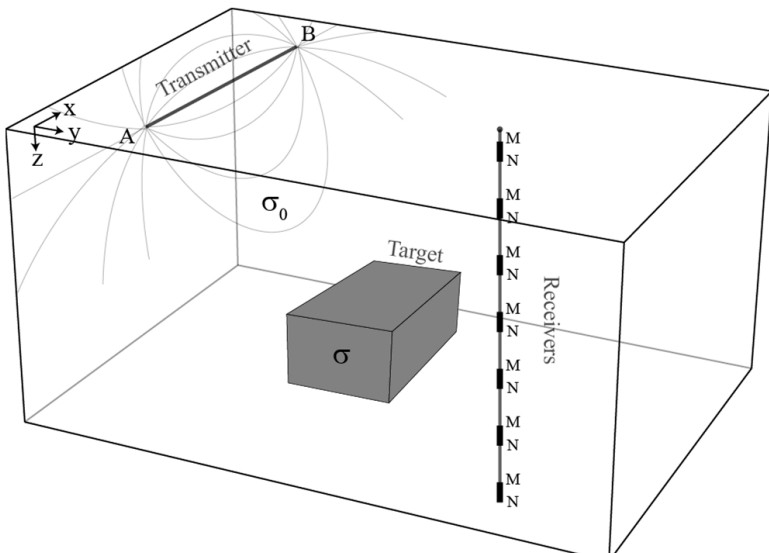

**Figure 1.** SBTEM survey configuration.

### 2.1. Regularized Inversion

Since 3D SBTEM inversion is nonlinear and underdetermined, we use regularization to solve the problem and construct the objective function based on an L2 norm measure, i.e.,:

$$\varphi(\mathbf{m}, \mathbf{d}_{\mathrm{pre}}) = \|\mathbf{W_d}(\mathbf{d}_{\mathrm{obs}} - \mathbf{d}_{\mathrm{pre}})\|_2^2 + \lambda \|\mathbf{W_m}(\mathbf{m} - \mathbf{m}_0)\|_2^2, \tag{1}$$

where $\mathbf{d}_{\mathrm{obs}}$ denotes the observed data of the $N_{\mathrm{d}} \times n_{\mathrm{channel}}$ dimension, while $\mathbf{d}_{\mathrm{pre}}$ denotes the predicted data from the inversion model. $N_{\mathrm{d}}$ and $n_{\mathrm{channel}}$ denote the number of receivers

and time channels. Here, we use the forward modeling code from Wang et al. [24] to calculate EM responses for the vector $\mathbf{d}_{\text{pre}}$. To better introduce the inversion scheme, in Appendix A, we give a simple introduction to the forward modeling algorithm based on the FE method using tetrahedral grids. $\lambda$ is a regularization factor, $\mathbf{W_d}$ is the data variance matrix with its elements being the reciprocal of the standard noise deviation, $\mathbf{W_m}$ is the model roughness operator, while $\mathbf{m}_0$ and $\mathbf{m}$ are the $M$-dimensional vectors, respectively, for the reference model and the solution.

To obtain the solution $\mathbf{m}^n$ of $n$th, we differentiate both sides of Equation (1) with respect to the model parameters $\mathbf{m}^{n-1}$ and obtain the gradient of the objective function, i.e.,:

$$\mathbf{g}^n = -2\mathbf{J}^{\mathrm{T}}\mathbf{W_d^T}\mathbf{W_d}\mathbf{r}^{n-1} + 2\lambda\mathbf{W_m^T}\mathbf{W_m}(\mathbf{m}^{n-1} - \mathbf{m}_0), \tag{2}$$

where $\mathbf{J}$ denotes the sensitivity matrix, and $\mathbf{r}^{n-1} = \mathbf{d}_{\text{obs}} - \mathbf{d}_{\text{pre}}^{n-1}$.

The sensitivity matrix in above equation is calculated by the adjoint forward modeling for fast calculation and reduction in memory consumption [25,26]. We assume that the predicted data satisfy $\mathbf{d} = \mathbf{Le}$, where $\mathbf{L}$ denotes a spatial interpolation operator. For the forward modeling in time-domain, $\mathbf{L}$ can be written as $\mathbf{L} = \mathbf{L_t}\mathbf{L_d}$, where $\mathbf{L_d}$ is the interpolation operator that interpolates the solution of the electric field to the receiver locations, while $\mathbf{L_t}$ is the interpolation operator that interpolates the solution of the calculated time channels to the observation ones. From the forward modeling scheme and $\mathbf{d} = \mathbf{Le}$, the sensitivity of EM responses with respect to the conductivity of the $k$th tetrahedral element, namely the element of the $k$th column in the overall sensitivity matrix, can be written as:

$$\mathbf{j}^k = \frac{\partial \mathbf{d}}{\partial m_k} = \frac{\partial \mathbf{Le}}{\partial m_k} = \mathbf{LK}^{-1}\left(\frac{\partial \mathbf{s}_{\text{TEM}}}{\partial m_k} - \frac{\partial \mathbf{K}}{\partial m_k}\mathbf{e}\right), k = 1, 2, \ldots, M. \tag{3}$$

Define a matrix $\mathbf{G}$ of the dimension $N \times M$ ($N = N_{edge} \times n_{channel}$) as:

$$\mathbf{G} = \left\{\frac{\partial \mathbf{s}_{\text{TEM}}}{\partial m_1} - \frac{\partial \mathbf{K}}{\partial m_1}\mathbf{e}, \frac{\partial \mathbf{s}_{\text{TEM}}}{\partial m_2} - \frac{\partial \mathbf{K}}{\partial m_2}\mathbf{e}, \ldots, \frac{\partial \mathbf{s}_{\text{TEM}}}{\partial m_M} - \frac{\partial \mathbf{K}}{\partial m_M}\mathbf{e}\right\}, \tag{4}$$

and substitute Equation (4) into (3); we obtain:

$$\mathbf{J}^{\mathrm{T}} = \mathbf{G}^{\mathrm{T}}\left(\mathbf{K}^{-1}\right)^{\mathrm{T}}\mathbf{L}^{\mathrm{T}}. \tag{5}$$

Equation (2) shows that to obtain the gradient of the objective function, we need to calculate the product of the transpose of the sensitivity matrix and a vector. Assuming that the multiplication vector is $\mathbf{v}$, then multiplying both sides of Equation (5) with $\mathbf{v}$ and defining $\mathbf{u} = \left(\mathbf{K}^{-1}\right)^{\mathrm{T}}\mathbf{L}^{\mathrm{T}}\mathbf{v}$, we obtain the following equations for the adjoint forward modeling, i.e.,:

$$\mathbf{K}^{\mathrm{T}}\mathbf{u} = \mathbf{L}^{\mathrm{T}}\mathbf{v}, \tag{6}$$

where the vector $\mathbf{v}$ is the term after the sensitivity matrix $\mathbf{J}$ in Equation (2). After solving the adjoint forward equation, we obtain the vector $\mathbf{u}$. Then, the product of the transpose of the sensitivity matrix $\mathbf{J}$ and the vector $\mathbf{v}$ can be calculated by:

$$\mathbf{J}^{\mathrm{T}}\mathbf{v} = \mathbf{G}^{\mathrm{T}}\mathbf{u}. \tag{7}$$

When calculating the matrix $\mathbf{G}$ in Equation (4), the term of $\frac{\partial \mathbf{K}}{\partial m_k}\mathbf{e}$ is analytical. As for the term of $\frac{\partial \mathbf{s}_{\text{TEM}}}{\partial m_k}$, the derivative of the source $\frac{\partial \mathbf{s}_i}{\partial m_k}$ is zero; the rest term can be written as:

$$\frac{\partial \mathbf{A}\mathbf{e}_0}{\partial m_k} = \mathbf{A}\frac{\partial \mathbf{e}_0}{\partial m_k} + \frac{\partial \mathbf{A}}{\partial m_k}\mathbf{e}_0, \tag{8}$$

where $\frac{\partial \mathbf{A}}{\partial m_k}\mathbf{e}_0$ can also be analytically calculated, while the term $\mathbf{A}\frac{\partial \mathbf{e}_0}{\partial m_k}$ can be calculated by adjoint method when executing the forward modeling for the DC field [25].

Since we use the unstructured tetrahedral mesh to discretize the calculation domain, the conventional model roughness operators for structured girds are not suitable. Thus, we modify the model roughness operator proposed by Key [27] for 2D problems to our 3D case [28] and obtain:

$$\|\mathbf{W_m}(\mathbf{m} - \mathbf{m}_0)\|^2 = \sum_{i=1}^{M} V_i \left[ \sum_{j=1}^{N(i)} \omega_j \left( \frac{\Delta m_{ij}}{\Delta r_{ij}} \right)^2 \right], \tag{9}$$

where

$$\Delta m_{ij} = m_i - m_j, \tag{10}$$

$$\Delta r_{ij} = \sqrt{\left(x_i - x_j\right)^2 + \left(y_i - y_j\right)^2 + \left(z_i - z_j\right)^2}, \tag{11}$$

$$\omega_j = \frac{V_j}{\sum\limits_{k=1}^{N(i)} V_k} \tag{12}$$

In the above equations, $V_i$ denotes the volume of the $i$th tetrahedron, while $N(i)$ denotes the number of the tetrahedrons that share vertices with the $i$th tetrahedron. The distance $\Delta r_{ij}$ between tetrahedrons is calculated from the centroid of each element.

### 2.2. Dual-Mesh Inversion Strategy

In the practical SBTEM survey, the boreholes are sparsely distributed in the working area. In most cases, only one borehole can be used. On the other hand, in 3D EM inversions, the areas close to the borehole receivers are generally refined to achieve accurate results. The sparse data combined with fine meshes can easily result in underdetermined 3D inversions. In this situation, an "equivalent trap" can frequently occur, namely the borehole data are well-fitted, but the model is not properly recovered. Under the condition of limited data, we can alternatively reduce the number of unknowns in the inversion by optimizing the mesh. In this paper, we propose a dual-mesh inversion strategy. First, we use a coarse mesh in the inversion to obtain rough structures in the underground. After that, we map the attributes in the coarse mesh to a fine one and continue the inversion. In this way, the inversion results from the coarse mesh can be taken as a good initial model, so that we can finally recover the underground structures in more detail. This can also help improve the data fitting in the inversions.

Referring to Figure 2, to do the mapping between the coarse and fine meshes, we take four basic situations into account: (1) the centroid of a fine mesh is inside a coarse one; (2) the centroid of a fine mesh is on the surface shared by two coarse meshes; (3) the centroid of a fine mesh is on the edge shared by several coarse meshes; and (4) the centroid of a fine mesh is at the vertex shared by several coarse meshes. For the first case, the attribute of the fine mesh is set to be equal to that of the coarse mesh that contains its centroid, while for all other cases we choose the coarse meshes related to the centroid of the fine mesh for calculating the attribute via volume-weighted average. Then, the attribute of the fine grid is calculated by:

$$(\sigma_i^e)_{\text{fine}} = \frac{\sum\limits_{j=1}^{m} \left(\sigma_{ij}^e\right)_{\text{coarse}}}{\sum\limits_{j=1}^{m} \left(\sigma_{ij}^e \cdot V_{ij}^e\right)_{\text{coarse}}}, i = 1, 2, \ldots, n, \tag{13}$$

where $\sigma_{ij}^e$ denotes the attribute of $i$th mesh. $m$ is the number of coarse meshes related to the centroid of the fine mesh, $n$ is the total number of fine meshes, and $\sigma_{ij}^e$ and $V_{ij}^e$ are the conductivity and volume, respectively, of the $j$th coarse mesh related to the centroid of $i$th fine mesh.

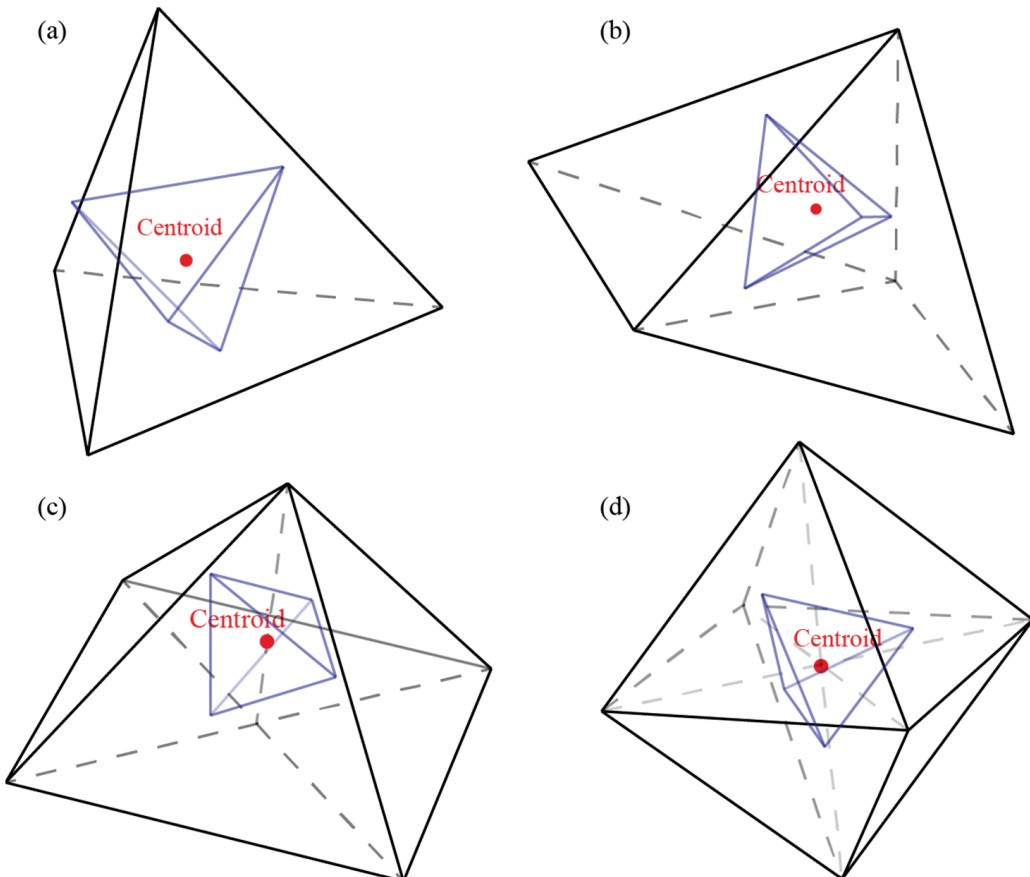

**Figure 2.** Schematic diagram of attribute mapping. The coarse mesh is in black, while the fine mesh is in blue. The centroid of the fine mesh is (**a**) within a coarse mesh, (**b**) on the surface shared by two coarse meshes, (**c**) at the edge shared by several coarse meshes, and (**d**) at the vertex shared by several coarse meshes.

## 3. Numerical Experiments

In this section, we will test our 3D SBTEM inversion scheme via both synthetic and field survey data. To remain consistent with the survey data, for all models in this section we adopt long wires as transmitting sources and assume the range of the calculation to be between $2.5 \times 10^{-4}$ s and $1 \times 10^{-2}$ s.

### 3.1. Flat Model Inversion

To validate our inversion algorithm, we first design a simple model with a block body buried in a uniform half-space (Figure 3). The two transmitting sources with the same length of 700 m are laid on the surface. Tx 1 is the transmitter corresponding to the ground receivers, while Tx 2 is the transmitter for the receivers in the borehole. The borehole receivers are located from 0 m to 600 m with an interval of 10 m. The borehole in the model is an open-hole well, which is hollow and full of air. The $E_z$ data in the borehole are measured by letting the electrodes touch the rock formation. To overcome the influence of limited data on our 3D inversion, we also put $31 \times 31 = 961$ survey stations at the earth surface with an interval of 20 m. The abnormal body's resistivity is 1 $\Omega \cdot$m, while the surrounding rock and the air is 100 $\Omega \cdot$m and $1.0 \times 10^8$ $\Omega \cdot$m, respectively. The forward model is discretized into 663,443 grids.

We perform 3D forward modeling to calculate $E_x$ on the surface and $E_z$ in the borehole and then added 3% Gaussian random noise to simulate the real situation. In the inversions, the model is first discretized into 562,888 grids as the coarse mesh, without refining elements close to the borehole and receivers. We use this mesh to do the inversion to the surface data, the borehole data, and the joint surface and borehole data, respectively. Figure 4

shows that the RMS for the surface data inversion is reduced to 0.998, but the other two inversions have difficulty achieving good data fittings. Next, we refine the elements close to the receivers and the borehole and discretize the model into 659,955 grids, and then we map the preliminary inversion results of the borehole data from the coarse mesh to the fine one. This operation will cause an RMS rising, but it does not obviously affect the final convergence. The final RMSs for the inversions of both the borehole data and the joint surface and borehole data reduce to the target level of 1.

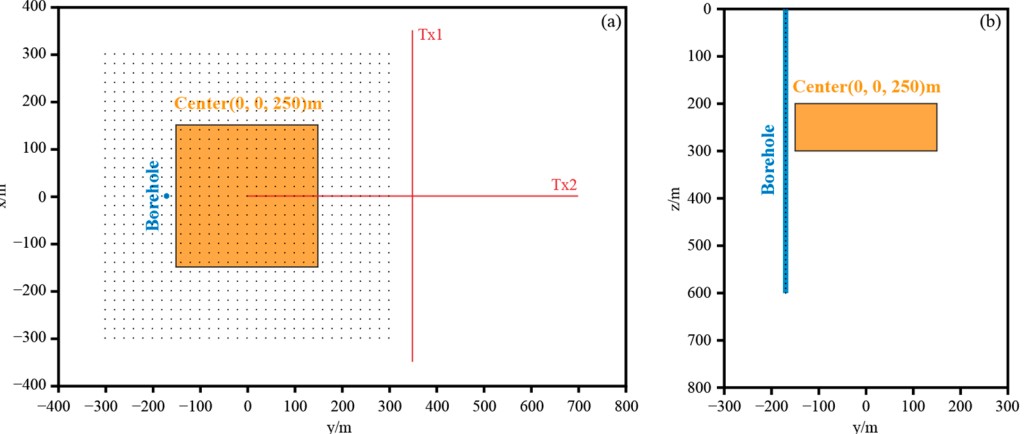

**Figure 3.** A synthetic model. (**a**) Plane view at $z$ = 250 m; (**b**) cross section at $x$ = 0 m. The orange blocks show the anomalous body. The two red lines denote the transmitting sources, while the blue dot at the left figure and the blue line at the right figure denote the borehole. The black dots at the left and right figures denote the receiver locations.

Figure 5 shows the inversion results using different strategies for different data. Figure 5a shows that the inversion only with the surface data roughly recover the model, but it has difficulty in recovering the boundaries of abnormal body, the recovered resistivity of the abnormal body is much larger than the true value. When we only invert the borehole data with fine mesh, since the sizes of the meshes near the receivers are very small, it is difficult to observe small meshes with drastic changes in attributes in plots of relatively large inversion areas. Figure 5b shows that the anomalous body is hardly to be observed as the inversion is too ill-posed. In this case, the borehole data can be easily fitted by a combination of resistivities of fine grids close to the receivers as they usually have large sensitivities. In contrast, when we use dual mesh for borehole data inversion, the anomalous body close to the borehole is well-inverted, especially its resistivity is well-recovered (Figure 5c). This succeeds because the coarse grids near the borehole cannot fit the data well, and thus, the inversion tries to find a model to fit the data by seeking the solution in a larger space of fine mesh. Figure 5d shows the joint inversion of surface and borehole data using fine mesh. The abnormal body is roughly recovered; however, the inverted body has a long tail in all sections. Finally, when we use the dual-mesh strategy to invert the joint surface and borehole data, from Figure 5e we can see that the best results are obtained; the boundaries and the resistivity of the abnormal body are well-recovered.

Summarizing the above discussion, we conclude that, by adding the surface data to SBTEM inversions, we can improve the resolution to the target around the borehole. Using a dual-mesh inversion strategy, we can improve the resolvability not only to the location but also to the resistivity of the target body.

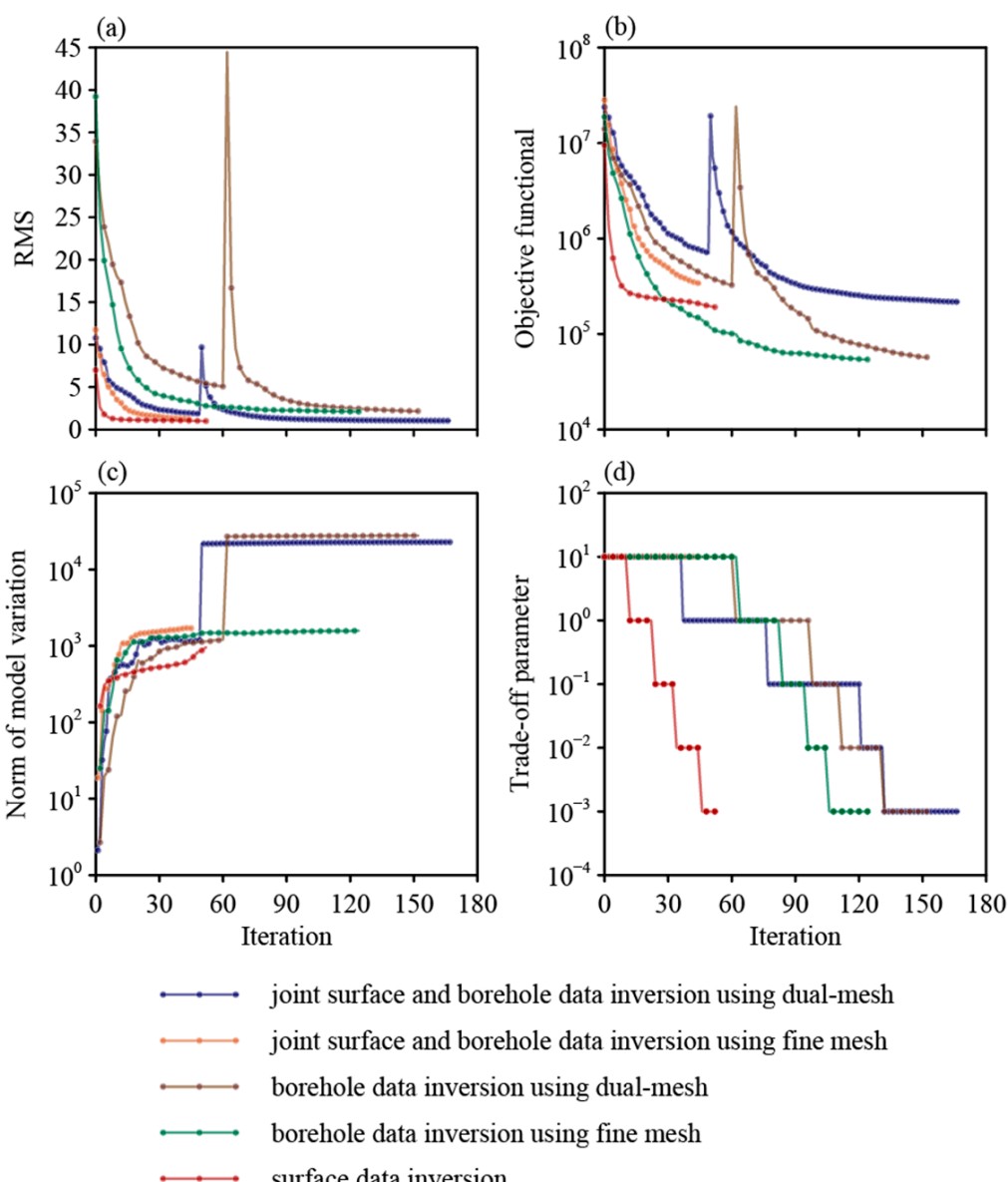

**Figure 4.** Inversion parameters versus iterations for the model in Figure 3. (**a**) RMS; (**b**) Objective functional; (**c**) Model; (**d**) Trade-off parameter.

### 3.2. Topographic Model Inversion

In this section, we further designed a synthetic model with real topography from a field dataset that will be illustrated in the next section. The locations of two grounded wire transmitters and data collection points at the surface are both taken from the field dataset. Referring to Figure 6, transmitting source 1 has a length of 2964 m, while transmitting source 2 has a length of 2791 m. The borehole is located at ($-600$ m, 0 m) with a depth of 600 m. The borehole receivers are laid at depths from 20 m to 600 m at an interval of 10 m. To remedy the limited data amount, we also take data acquired at the earth surface for our 3D inversion.

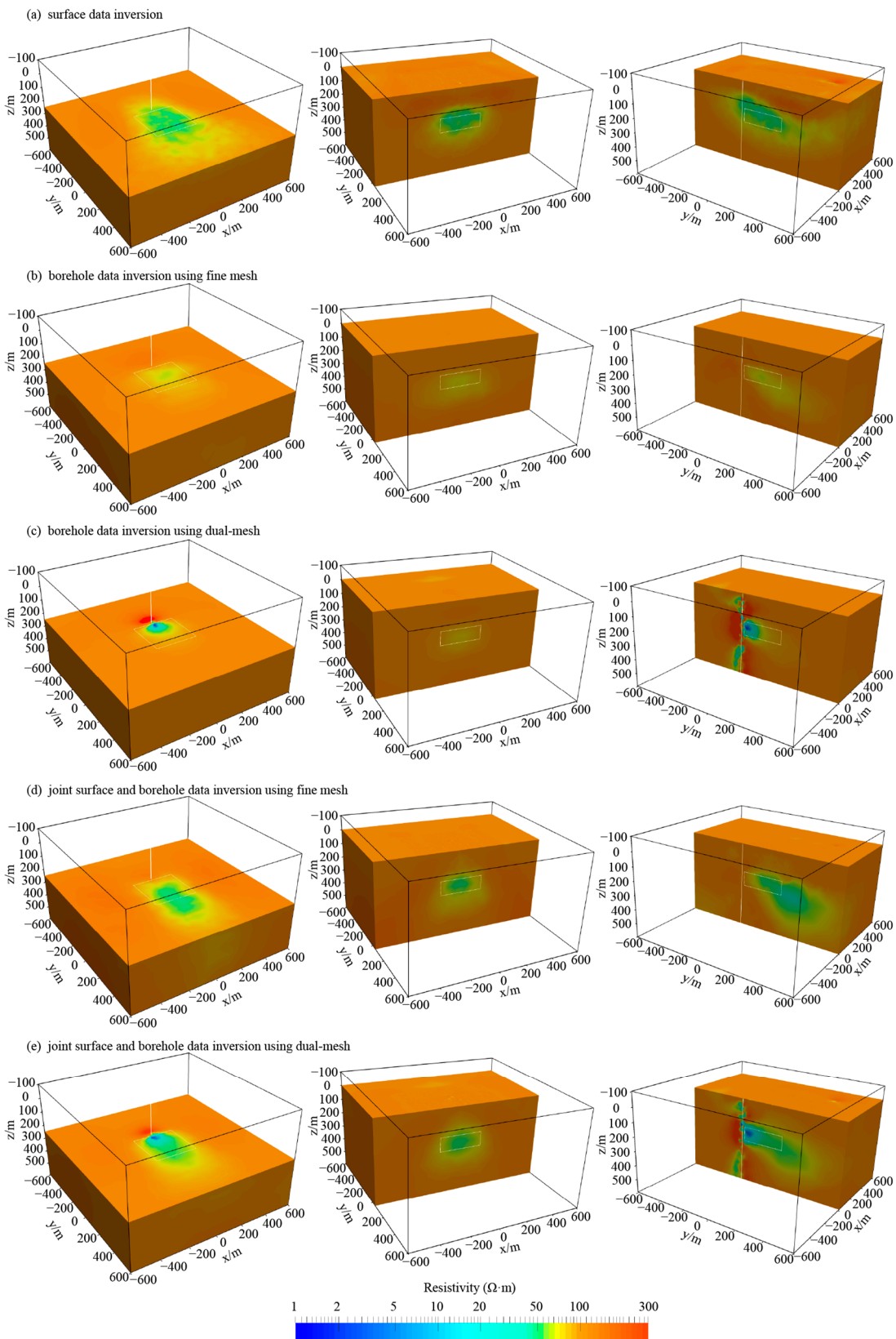

**Figure 5.** Inversions of the surface data, borehole data, and joint surface and borehole data for the model in Figure 3.

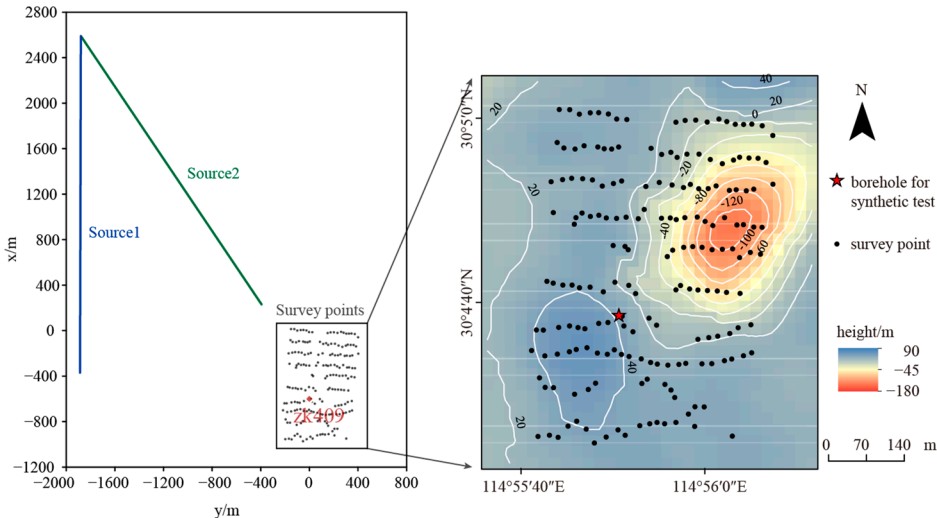

**Figure 6.** Source locations and survey points.

Referring to Figure 7, three conductive abnormal bodies with the resistivity of 1 Ω·m are embedded in the underground. The surrounding rock and the air have resistivities of 100 Ω·m and $1.0 \times 10^8$ Ω·m, respectively. The model is first discretized into 472,151 coarse grids and, then, into 664,173 fine grids. We perform 3D forward modeling on this synthetic model and compare the responses with those for a uniform half-space. From Figure 8, one sees that the conductive bodies cause obvious anomalies. Considering that the electric fields in the x- and y-directions are not easy to measure in the borehole, we use only the $E_z$ component as the data and add 10% Gaussian random noise to simulate the real situation. In our inversion process, we use a cooling schedule to determine regularization factor λ, namely when the data fitting decreases very slowly, the regularization factor becomes 0.1 times the previous value and remains unchanged after reaching a threshold.

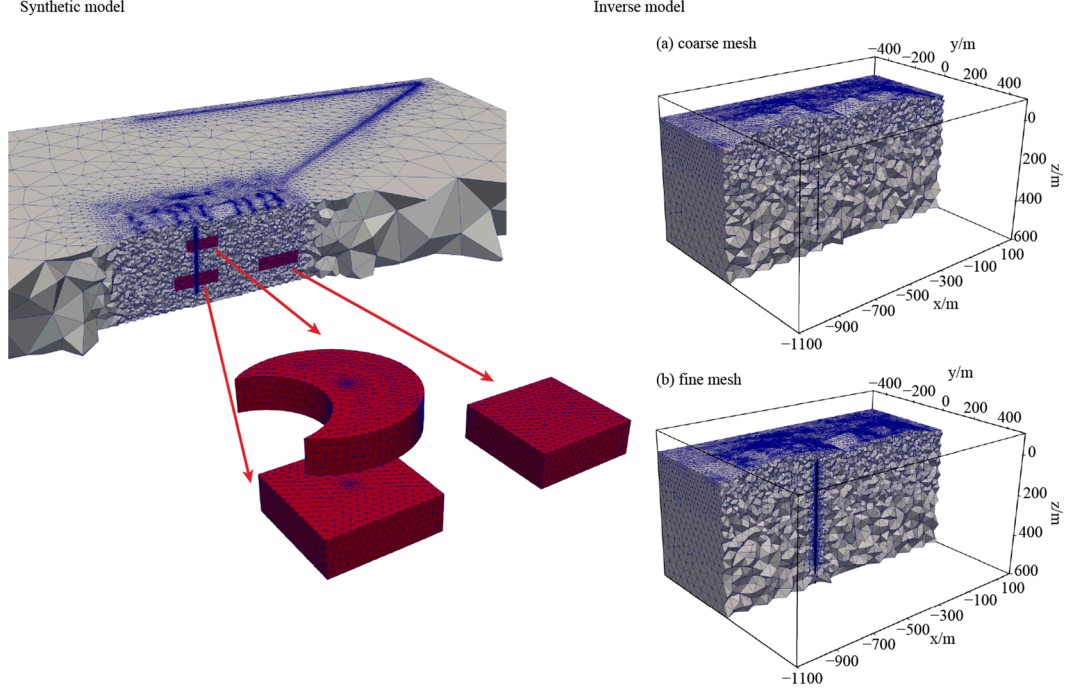

**Figure 7.** Three 3D abnormal bodies embedded in a topographic half-space.

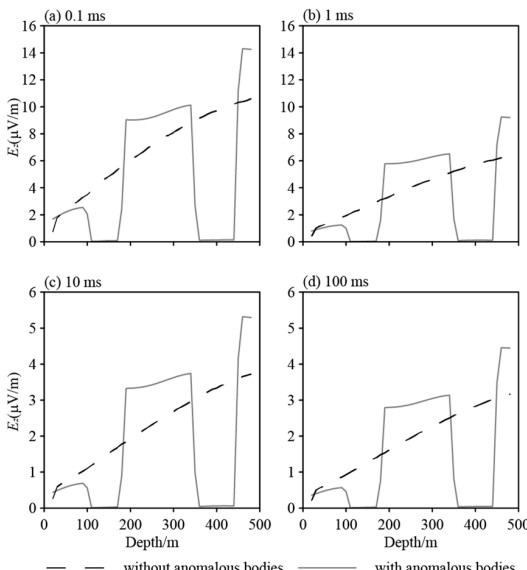

**Figure 8.** Responses of receivers in the borehole in different times for the synthetic model in Figure 7 with and without anomalous bodies.

To undertake our dual-mesh inversions, we first use the coarse mesh in Figure 7a to recover the preliminary model. After 25 iterations, we interpolate the inversion results to the fine grids in Figure 7b using the mapping scheme in Figure 2 and continue the inversion with the fine grids until the stop condition is satisfied (i.e., RMS ≤ 0.98). The inversion runs 6 h and 25 min on the coarse mesh with the RMS reduced to 5.05. After mapping to the fine mesh, the inversion runs again 16 h and 28 min on the fine mesh, the RMS is finally reduced to 0.98. In addition, we also run 3D inversions using the conventional single-mesh scheme with the fine mesh. The inversion takes 26 h and 37 min, with the RMS finally reduced to 0.97. Figure 9 compares the inversion parameters between our dual-mesh scheme and the conventional one. For both methods, the RMS and the objective function decrease rapidly at the early time and then get stabilized. Although the parameters for our dual-mesh inversions jump when mapping between two meshes, the overall RMS and objective function attenuate, and the inversion converges.

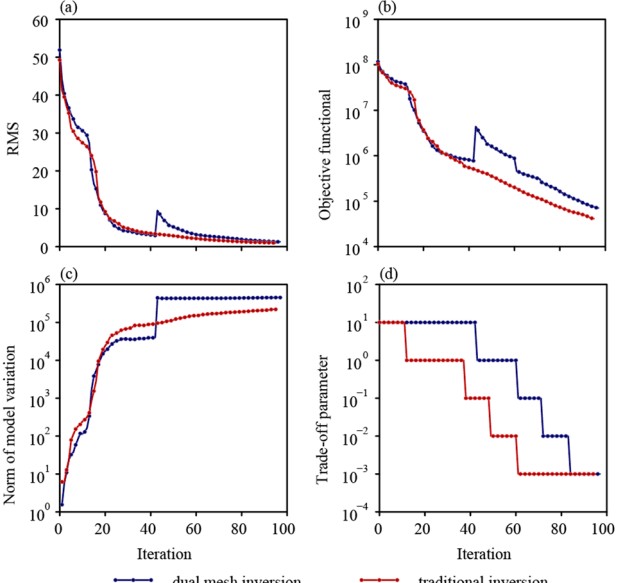

**Figure 9.** Inversion parameters versus iterations for the dual-mesh and conventional methods. (**a**) RMS; (**b**) Objective functional; (**c**) Model; (**d**) Trade-off parameter.

Figure 10 shows the comparison of inversion results from the two methods. The conventional inversion has difficulty in recovering the conductive anomalous bodies, while our dual-mesh inversion can well-recover the target bodies. The reason for this difference is that when we do the conventional inversion, we invert all underground structures simultaneously. For very limited data, the inversion is heavily dependent on the initial model and can easily be trapped in local minima. At this time, although the data are well-fitted, the recovered model may be far away from the true one (the so-called "equivalent trap"). In contrast, using our dual-mesh scheme, we invert the data first for rough but main underground structures, and then, we map them to the fine mesh as the starting model. Considering that our inversion on the fine mesh keeps the main underground structures unchanged while modifying the local ones, our method cannot be easily trapped in local minima. This inversion scheme follows the basic concept of multiscale inversions. From the data fitting for the two inversions shown in Figure 11, one sees that for the dual-mesh inversion, the data fitting of the inversion with coarse mesh is not good. After the inversion with fine mesh, however, the data fitting becomes as good as the single-mesh inversion. From the data fitting at the earth surface shown in Figure 12, we can also see that the data at the ground surface are well-fitted in both inversions.

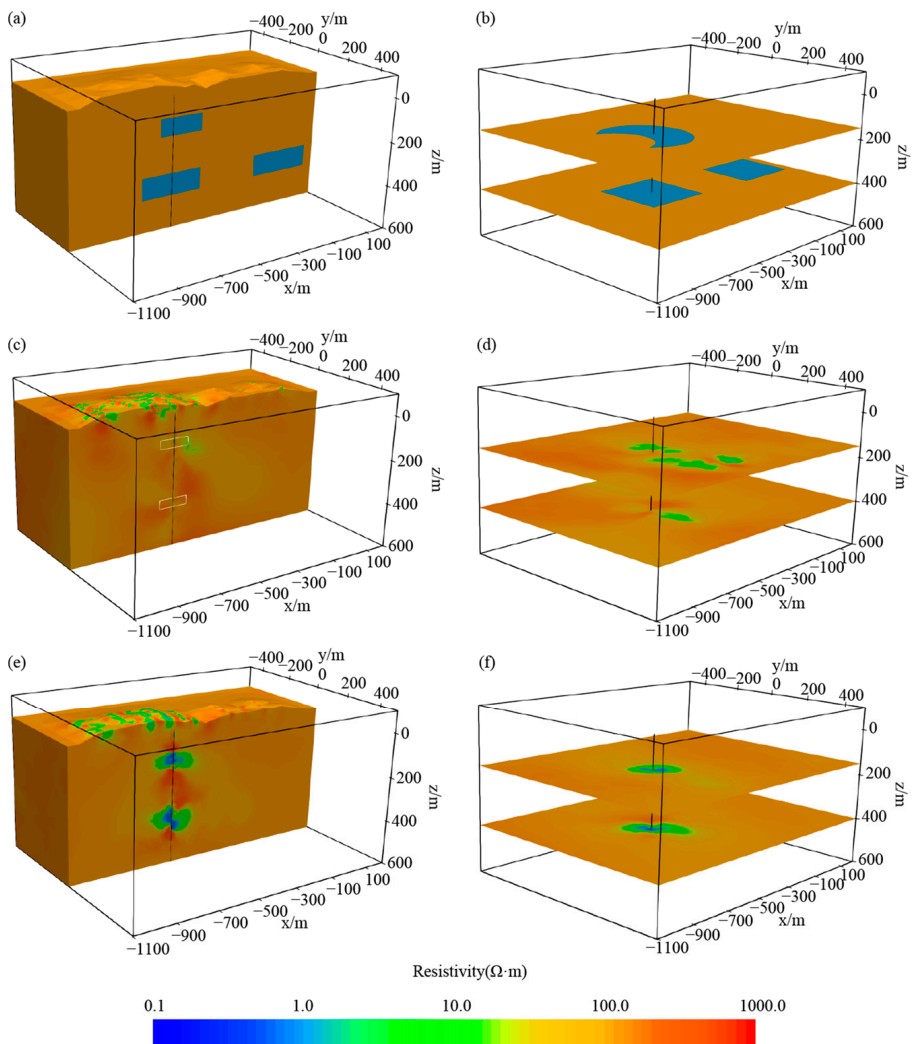

**Figure 10.** True model and inversion results from the dual-mesh and conventional methods: (**a**,**b**) true model; (**c**,**d**) conventional inversions; (**e**,**f**) dual-mesh inversions.

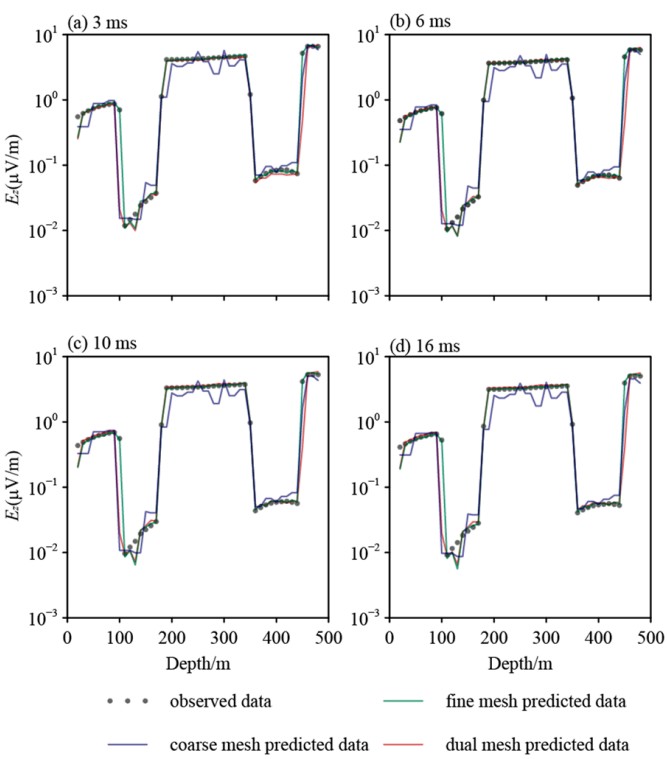

**Figure 11.** Borehole data fitting for dual-mesh and conventional methods.

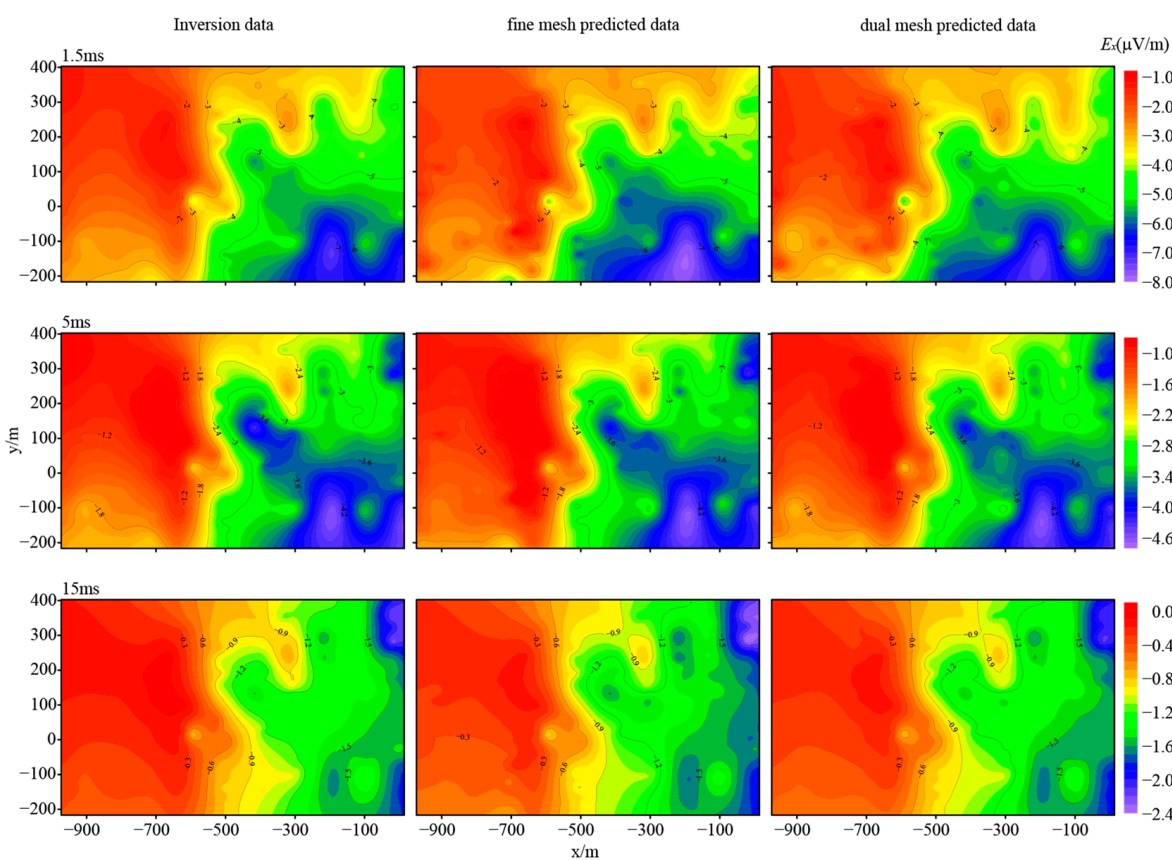

**Figure 12.** Data fitting at the earth surface for dual-mesh and conventional methods.

*3.3. Field Data Inversion*

In this section, we take a dataset acquired from the Western Tonglu Mountain copper mine in Daye, China to test the effectiveness of our dual-mesh method. The lithology in the survey area are as follows. From 0 to 727.16 m, it is mainly quartz monzodiorite porphyrite; from 727.16 to 1031.80 m, it is marble with an embedded layer of copper and an iron ore body at the depth of 763.16–778.86 m. This ore body is controlled by the upper contact zone of low resistivity. From 1031.8 to 1036.3 m, it is skarn, while from 1036.3 to 1107.2 m, it is quartz monzodiorite. The copper-bearing hematite and magnetite located at the depth of 1040.4–1046.95 m are controlled by the lower contact zone. Compared to the upper contact zone, the contents of chalcopyrite and magnetite in the lower contact zone are significantly reduced, so that we can see the low-resistivity characteristics in this area.

Referring to Figure 6, the borehole ZK409 was drilled for SBTEM exploration. The depth of ZK409 is 1107.2 m. The positions of transmitting source and survey points are also shown in Figure 6. The geodetic coordinates of the starting and ending points for Source 1 are (3,332,207.8, 385,87,811.2) and (3,329,244.3, 38,587,804.4), while for Source 2 they are (3,332,207.8, 38,587,811.2) and (3,329,845.9, 38,589,298.8). The borehole ZK409 has coordinates of (3,329,616.93, 38,589,691.10).

Figure 13 shows the dual meshes used in our inversions. As we only observe the borehole data within the depth range of 600–900 m, we discretize this section into fine grids. We take the ground $E_x$ and the borehole $E_z$ as the inversion data and assume 5% of amplitudes as the noise floor. The air resistivity is set to $1 \times 10^8$ Ω·m, while the background resistivity is assumed to be 500 Ω·m. The regularization factor is selected following the same strategy as in the synthetic examples. The inversion for the coarse mesh runs 12 h and 51 min, and the RMS decreases to 11.8. After mapping to the fine mesh, the inversion runs again 52 h and 28 min, and the RMS decreases to 4.43. Figures 14 and 15 show the data fitting at the earth surface and in the borehole. The surface data are well-fitted, while for the borehole data, the electric field $E_z$ is overall well-fitted, except for the depth range of 620–720 m at early time channels. The good data fitting implies that the inversion using our dual-mesh scheme converges.

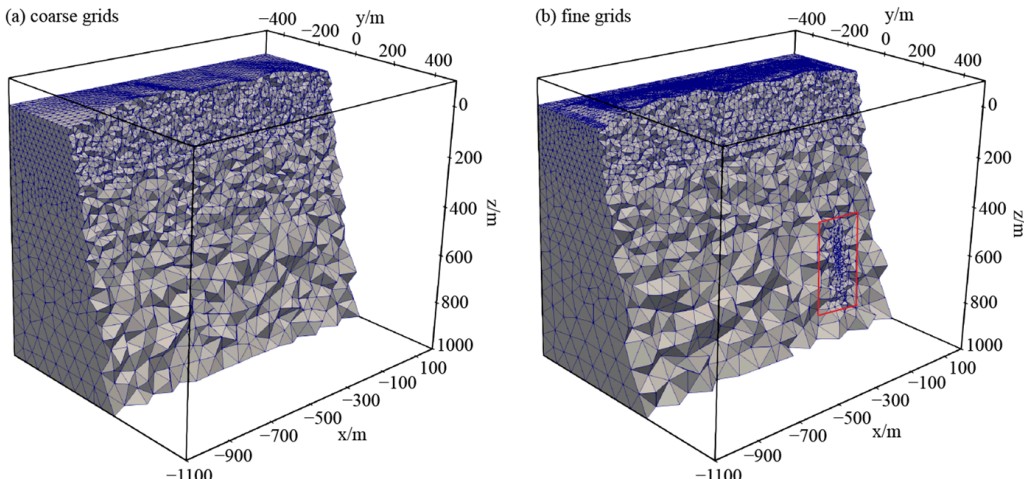

**Figure 13.** Meshes for our dual-mesh inversion of survey data from Daye, China.

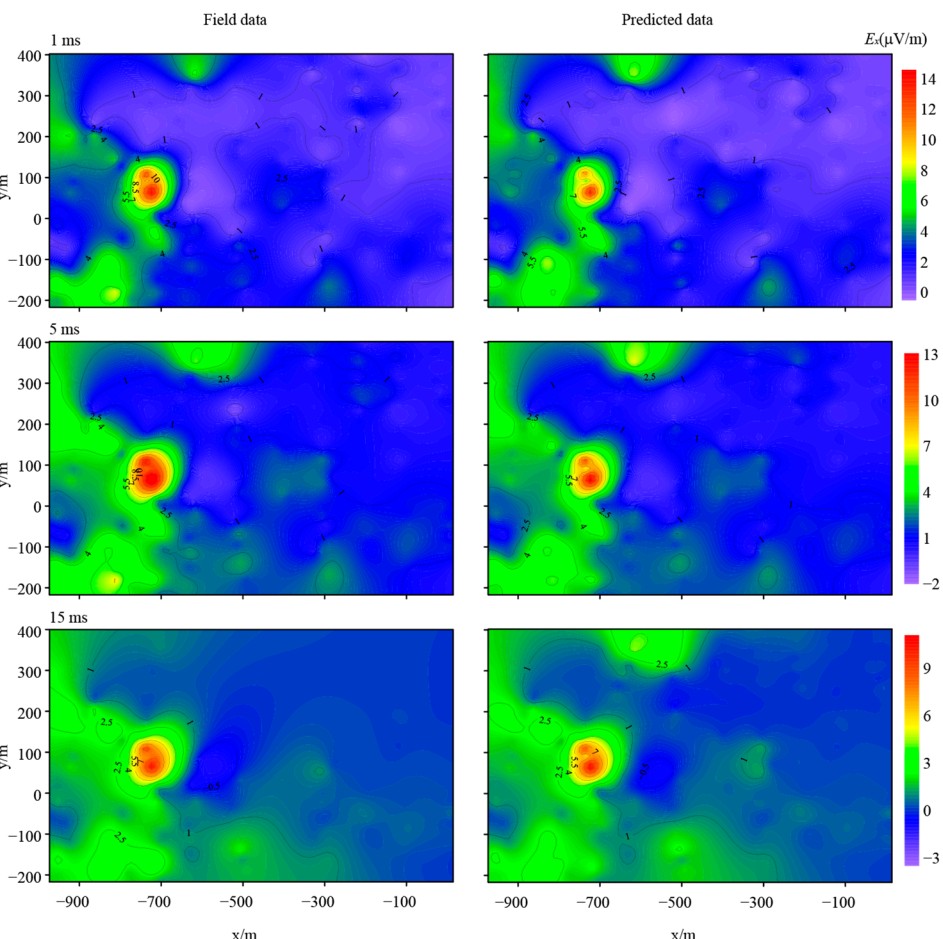

**Figure 14.** Surface data fitting for the field data inversion.

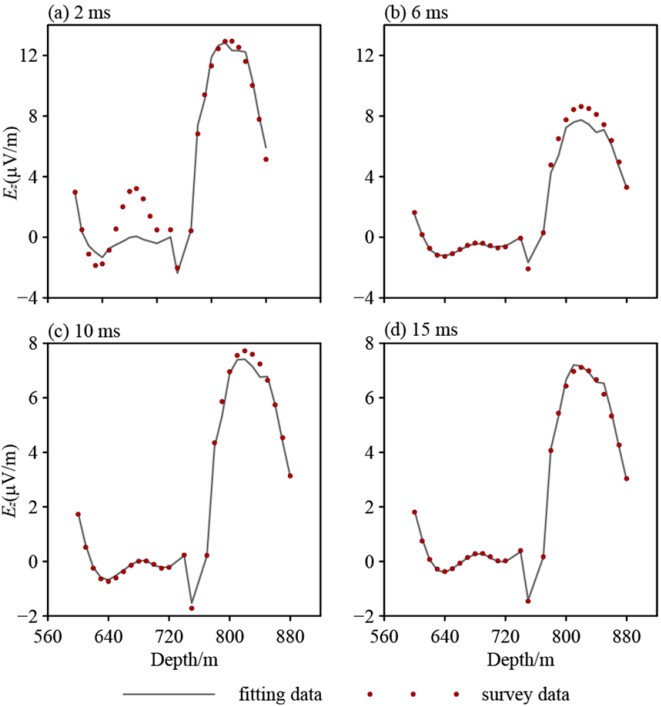

**Figure 15.** Borehole data fitting for the field data inversion.

Figure 16 shows the inversion results of the survey data from the Western Tonglu Mountain copper mine. The low-resistivity anomalous body at 650–770 m is clearly revealed with a minimum resistivity of about 5 Ω·m that is consistent with the resistivity of the rock sample in Table 1 (mainly the copper, iron ore body). The recovered resistivity of the surrounding rock is from 300 to 1000 Ω·m, which is consistent with the high-resistivity quartz monzodiorite porphyrite with copper mineralization or middle-resistivity dolomitic marble and marble with copper mineralization. The logging data show that there exist two low-resistivity zones, respectively located at the depth ranges 70–340 m and 660–800 m, corresponding well to the locations of the conductive anomalies revealed in Figure 16. It should be pointed out that since, from the borehole survey, we only obtain the data at the depth range of 600–900 m, we are not able to clearly characterize the low-resistivity anomalies outside this range.

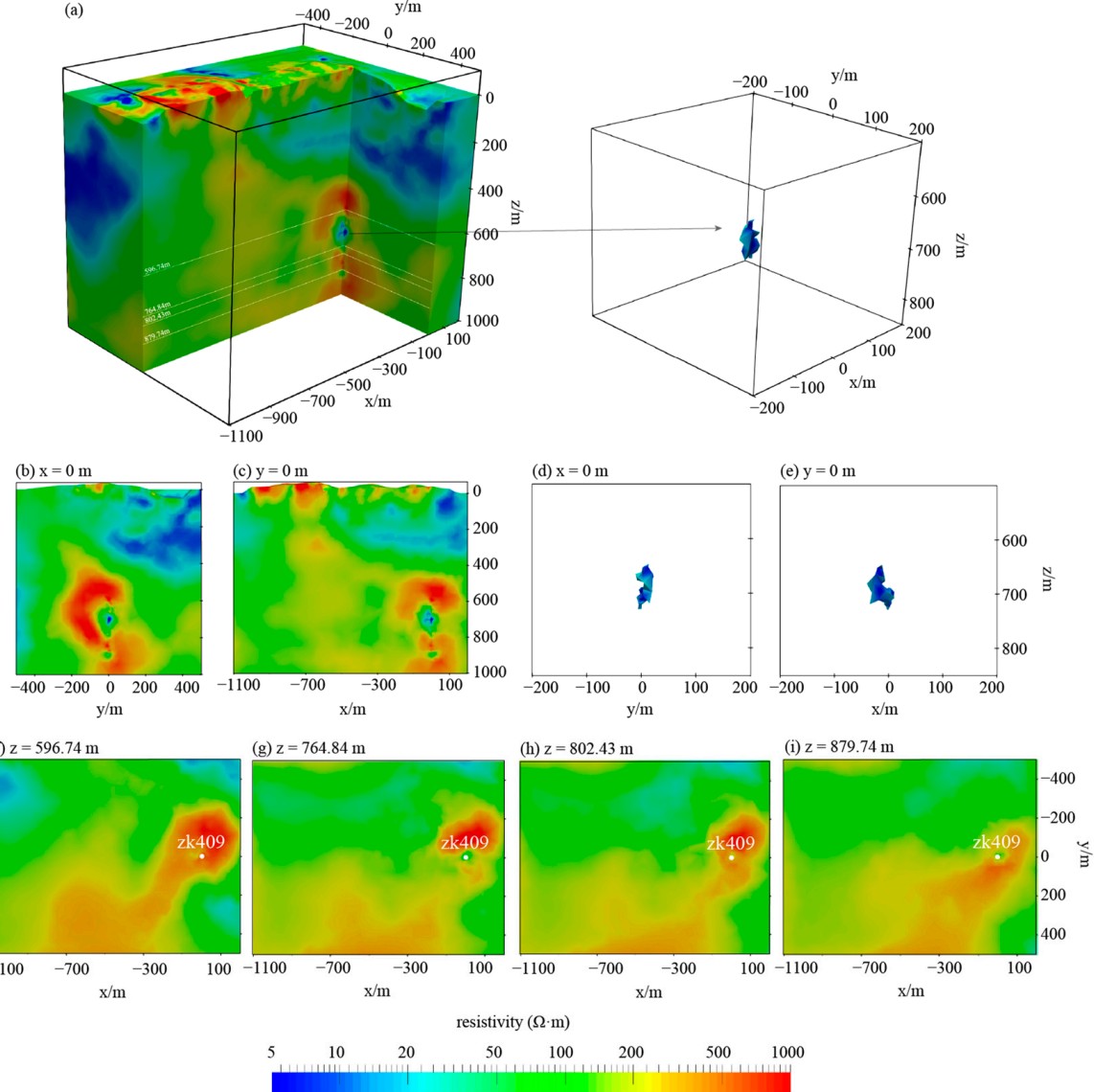

**Figure 16.** Dual-mesh inversions of the survey data from the Western Tonglu Mountain copper mine in Daye, China. (**a**) 3D resistivity distribution; (**b**) slice at *x* = 0 m; (**c**) slice at *y* = 0 m; (**d**) ore body in borehole at *x* = 0 m; (**e**) ore body in borehole at *y* = 0 m; (**f**–**i**) *x*,*y* slices at rock sample positions.

**Table 1.** Resistivities of rock samples from borehole ZK409.

| No. | Borehole Depth (m) | Core Properties | $\rho_0$ ($\Omega \cdot m$) |
|---|---|---|---|
| wx-1 | 596.74–596.84 | porphyritic quartz monzodiorite | 7178 |
| wx-2 | 764.84–764.94 | copper-bearing hematite magnetite ore | 8.7 |
| wx-3 | 802.43–802.54 | marble rock | 1125 |
| wx-4 | 879.75–879.85 | marble rock | 374 |

In Figure 16f–i, we plot four slices corresponding to the location of rock sampling. The rock sampling and the inversion results show relatively high-resistivity characteristics at $z = 596.74$ m and $z = 802.43$ m, while the rock sampling and the inversion results at $z = 764.84$ m and $z = 879.74$ m show relatively low-resistivity characteristics. The low-resistivity anomaly at $z = 764.84$ m is our target.

## 4. Discussion

The advantage of the SBTEM method is that the receivers are located in the underground near the abnormal body, so it can provide much higher resolution for a deep target than the conventional method. However, due to limited boreholes, when we use traditional dense local meshes (near the borehole) for inversion, the ambiguity is very strong, as can be seen from Figure 16b,c. In contrast, our dual-mesh inversion has distinctive advantages: (1) we use the coarse mesh to invert the data that can outline the main structures in the underground; (2) then, we use the fine mesh to invert the fine structure and obtain good data fitting. In this way, we can get good recovery to the underground structure and the good data fitting, and the uniqueness of the inversion can be improved.

Although the dual-mesh strategy overcomes ambiguity for the SBTEM data inversion to some extent, the limited resolution of the borehole data still cannot be solved. Only the targets close to the borehole receivers can be recovered, while the others cannot be restored. By jointly inverting the borehole and surface data, as can be seen from Figure 16c,e, the inversion result can be improved for the targets close to the borehole, but it does not work for the targets far away from the borehole.

It needs to be pointed out that the induced polarization (IP) effect is often observed when TEM with grounded lines is applied in the field, especially for massive sulfide or graphite exploration. However, from the rock-physics experiments on the samples collected from the borehole in our survey area, we did not see an obvious large IP effect; thus, we did not consider it in our 3D inversions. Considering that an IP effect can seriously affect the inversion results of TEM data, special attention needs to be paid in areas with an obviously strong IP effect.

## 5. Conclusions

Based on the unstructured finite-element method and back Euler discretization, we successfully developed a dual-mesh scheme for SBTEM data inversions. Numerical experiments for synthetic models showed that by adding the surface data to SBTEM inversions, we can get better results than when only the borehole data are used in the inversion. By taking the dual-mesh strategy, we can further improve the resolutions of both the target location and the resistivity. More inversions of synthetic data with multiple bodies under topographic earth showed that in contrast to the conventional methods that invert the data once, our dual-mesh method inverts the data in a multiscale way that can suppress the non-uniqueness of the SBTEM inversions to some extent. The inversion of the field dataset from the Western Tonglu Mountain copper mine also demonstrated that our dual-mesh method can deliver good results. The inversions are in good agreement with the resistivity measurements of the borehole rock samples and well logging data.

**Author Contributions:** Conceptualization, L.W., Y.L. and C.Y.; methodology, L.W., C.Y. and Y.L.; software, L.W. and Y.L.; formal analysis, L.W. and Y.L.; investigation, L.W., C.Y., Y.L., B.Z., X.R. and Y.S.; writing—original draft preparation, L.W.; writing—review and editing, L.W. and C.Y.; funding acquisition, C.Y., Y.L., B.Z. and X.R. All authors have read and agreed to the published version of the manuscript.

**Funding:** This paper is financially supported by the National Key Research and Development Program of China (2021YFB3202104) and the National Natural Science Foundation of China (42030806, 42074120, 42174167, 42274093).

**Data Availability Statement:** Data associated with this research are available and can be obtained by contacting the corresponding author.

**Conflicts of Interest:** The authors declare no conflict of interest.

## Appendix A

In the appendix, we will introduce how to calculate the SBTEM responses using the finite-element method. The governing equation for SBTEM forward modeling can be obtained from the following curl–curl equations, i.e.,:

$$\nabla \times \left[ \frac{1}{\mu_0} \nabla \times \mathbf{E}(\mathbf{r}, t) \right] + \sigma \frac{\partial \mathbf{E}(\mathbf{r}, t)}{\partial t} + \frac{\partial \mathbf{j}_s(\mathbf{r}, t)}{\partial t} = 0, \tag{A1}$$

where $\mathbf{E}(\mathbf{r},t)$ is the electric field, $\mu_0$ is the permeability of the vacuum, $\sigma$ is the conductivity, and $\mathbf{j}_s(\mathbf{r},t)$ is the source current.

We first use the open-source code TetGen [29] to discretize the model domain into unstructured tetrahedral elements. Referring to Jin [30], for our forward modeling based on the finite-element method, we write the electric field at time $t$ and location $\mathbf{r}$ in an interpolation format as $\mathbf{E}^{\mathrm{e}}(\mathbf{r}, t) = \sum_{j=1}^{6} e_j^{\mathrm{e}}(t) \mathbf{n}_j^{\mathrm{e}}(\mathbf{r})$, where $e_j{}^{\mathrm{e}}(t)$ denotes the electric field at the $j$th edge, while $\mathbf{n}_j{}^{\mathrm{e}}$ denotes the vector interpolation basis functions. Inserting this interpolation into Equation (A1) and multiplying both sides by a weighting function (it usually takes the basis function) and integrating over the element, we get:

$$\mathbf{A}^{\mathrm{e}} \frac{\mathrm{d}\mathbf{e}^{\mathrm{e}}(t)}{\mathrm{d}t} + \mathbf{B}^{\mathrm{e}} \mathbf{e}^{\mathrm{e}}(t) + \mathbf{s}^{\mathrm{e}} = 0, \tag{A2}$$

where $\mathbf{e}^{\mathrm{e}} = \left[ e_1^{\mathrm{e}}, e_2^{\mathrm{e}}, \dots, e_n^{\mathrm{e}} \right]$ are the electric fields at the element edges. The mass matrix $\mathbf{A}^{\mathrm{e}}$, the stiffness matrix $\mathbf{B}^{\mathrm{e}}$, and the source term $\mathbf{s}^{\mathrm{e}}$ can be written as:

$$A_{ij}^{\mathrm{e}} = \iiint_{V^{\mathrm{e}}} \sigma^{\mathrm{e}} \mathbf{n}_i^{\mathrm{e}}(\mathbf{r}) \cdot \mathbf{n}_j^{\mathrm{e}}(\mathbf{r}) \mathrm{d}V, \tag{A3}$$

$$B_{ij}^{\mathrm{e}} = \iiint_{V^{\mathrm{e}}} \nabla \times \mathbf{n}_i^{\mathrm{e}}(\mathbf{r}) \cdot \nabla \times \mathbf{n}_j^{\mathrm{e}}(\mathbf{r}) \mathrm{d}V, \tag{A4}$$

$$s_i^{\mathrm{e}} = \iiint_{V^{\mathrm{e}}} \mathbf{n}_i^{\mathrm{e}}(\mathbf{r}) \cdot \frac{\partial \mathbf{j}_s(\mathbf{r}, t)}{\partial t} \mathrm{d}V. \tag{A5}$$

To solve Equation (A2), we adopt the implicit backward Euler scheme for time discretization, which is unconditionally stable [19,31]. For that purpose, we divide the time channels into several ranges and assume a small step for the early time but a large step for the later time. The initial time step is chosen to be 1/100 of the first time channel, and the rest time step is twice the previous time step.

Assembling the element matrix (A3) and (A4) into a global one and substituting it into Equation (A2), we obtain the following recursive formulation for the electric field, i.e.,:

$$\mathbf{C}\mathbf{e}^{n+2} = \mathbf{A}\left( 4\mathbf{e}^{n+1} - \mathbf{e}^{n} \right) - 2\Delta t \mathbf{s}^{n+2}, \tag{A6}$$

where $\mathbf{C} = 3\mathbf{A} + 2\Delta t\mathbf{B}$. The final equation for the solution of the electric field can be written as:

$$
\begin{bmatrix}
\mathbf{C}_1 & & & & & & \\
-4\mathbf{A} & \mathbf{C}_2 & & & & & \\
\mathbf{A} & -4\mathbf{A} & \mathbf{C}_3 & & & & \\
& \mathbf{A} & -4\mathbf{A} & \mathbf{C}_4 & & & \\
& & \ddots & \ddots & \ddots & \\
& & & \mathbf{A} & -4\mathbf{A} & \mathbf{C}_n
\end{bmatrix}
\begin{bmatrix}
\mathbf{e}_1 \\ \mathbf{e}_2 \\ \mathbf{e}_3 \\ \mathbf{e}_4 \\ \vdots \\ \mathbf{e}_n
\end{bmatrix}
=
\begin{bmatrix}
-2\Delta t_1\mathbf{s}_1 + 3\mathbf{A}\mathbf{e}_0 \\
-2\Delta t_2\mathbf{s}_2 - \mathbf{A}\mathbf{e}_0 \\
-2\Delta t_3\mathbf{s}_3 \\
-2\Delta t_4\mathbf{s}_4 \\
\vdots \\
-2\Delta t_n\mathbf{s}_n
\end{bmatrix},
\qquad (A7)
$$

where $\mathbf{e}_0$ represents the initial electric field.

Since we assume a grounded wire as the source to transmit a step-off current into the earth, we need to solve a Poisson's equation for the initial DC field. We use the nodal finite-element method to calculate potentials at the nodes and then calculate $\mathbf{e}_0$ by interpolations and spatial derivatives. As the distance from the source increases, the grid size gets bigger. We need to expand the calculation domain far away from the source, so that we can apply the homogeneous Dirichlet boundary condition by setting the electric field on the outer boundary to zero. Equation (A7) can be simplified to:

$$
\mathbf{K}\mathbf{e} = \mathbf{s}_{\text{TEM}}, \qquad (A8)
$$

where $\mathbf{K}$ is the coefficients matrix, and $\mathbf{s}_{\text{TEM}}$ is the source term. The direct solver MUMPS [32] is used to solve Equation (A8) for the electric field at each edge. Then, the electric field at the receiver locations can be calculated by interpolations, while the magnetic field can be calculated by Faraday's law.

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
