# Peer review of "Three-Dimensional Dual-Mesh Inversions for Sparse Surface-to-Borehole TEM Data"

_remotesensing, doi:10.3390/rs15071845_

Round 1
Reviewer 1 Report
Although this paper focuses more on numerical methods, still, this is a study dealing with physics problem of SBTEM. However, the authors jumped to the ‘Method’ (which is pure of mathematics) in section 2 without showing the model and description of physics. How can a reader understand the ‘method’ without physics background?
The model is finally shown in section 3 in Fig. 2. However, the Fig. 2 is not well explained.
I am afraid it will be very difficult for the readers to understand this work.
Author Response
Dear Reviewer:
Thank you very much for taking time to review our paper and providing useful suggestions. We have made revisions to the manuscripts according to your suggestions. Our detailed responses to your comments are listed as follows.
With best regards,
Changchun Yin

Reviewer 2 Report
Dear Authors,
I am glad to review your paper. This paper proposes the SBTEM inversion algorithm and applies 3D SBTEM inversion scheme to synthetic and field survey data. It is novel and interesting. This paper can be accepted after some minor improvement.
1) Introduction: "In past decades, the SBTEM has been widely used in mineral explorations. Lennart and Robert [3] …Paggi and Macklin[7] applied the SBTEM method to the exploration of Eureka Volcano vein." These references can be summarized into one or two sentences.
2)3.3. Field data inversion: The observed data may be affected by the induced polarized effect of the ore body or surrounding rocks.
Best regards
Author Response

(The authors gave the same response as above.)

Reviewer 3 Report
Overall the proposed dual meth algorithm for data inversion is an interesting approach. The paper uses both numerical and experimental data and have show good inversion results. However, that are few unanswered quesions, which need to addressed before publications:
1. What is frequency range, or what is the TEM wave form?
2. How what modeled wire? what was discretization size on wire?
3. do you consider skin depth in high conducting wire?
4. Is the model accurate? could you show comparisons against other methods?
additional comments are inserted in the attached paper.

Author Response
Dear Reviewer:
Thank you very much for taking time to review our paper and providing useful suggestions. We have made revisions to the manuscripts according to your suggestions. Our detailed responses to your comments are listed as follows.
In addition, we have also made annotations in the attached manuscript.
With best regards,
Changchun Yin

Reviewer 4 Report
The paper devoted to three-dimensional dual mesh inversions for sparse surface-to-borehole TEM data.
The scientific idea of paper is good and important for geophysicists who apply TEM data inversion all over the world.
In general the structure of paper is fine and clear, but Discussion chapter must be added.
Good introduction and the surface-to-borehole transient electromagnetic (SBTEM) method inversion background are given. Authors give details how do they regularize inversion mathematically, and that is nice! Appendix is also given. Dual mesh inversion strategy described in details as well.
But since the surface-to-borehole transient electromagnetic (SBTEM) method with grounded wire as transmitter is used, why the induced polarization (IP) effect is not considered? Assumed geoelectric model does not include chargeability properties of layers. But we always observe IP effect when TEM with grounded lines is applied in the field. Hence the workflow of modeling experiment can be uncertain and disputable. And this is the crucial point. Detailed explanation on this must be given by authors.
Authors use different levels of Gaussian noise to be used in modelling. Why the levels are different? Based on what assumption they were chosen?
Finally, authors pose that applied dual mesh inversion is better than conventional approach. But from the Figures (cubes and slices) it is not evident. Maybe authors should choose a bit more unpretentious manner of results interpretation? Anyway this should be explained in Discussion chapter.
Some minor corrections should be done according to the comments listed in attached PDF file.

Author Response
Dear Reviewer:
Thank you very much for taking time to review our paper and providing useful suggestions. We have changed the position of the drawings in the text to enhance the readability of the manuscript and made revisions to the manuscripts according to your suggestions. Our detailed responses to your comments are listed as follows.
In addition, we have also made annotations in the attached manuscript.
With best regards,
Changchun Yin

Round 2
Reviewer 1 Report
It is much improved.